# One-Step Synthesis of Cross-Linked Esterified Starch and Its Properties

**Xiaozhou Xue *** , **Qi Liang, Qunyu Gao *** and **Zhigang Luo**

School of Food Science and Engineering, South China University of Technology, Guangzhou 510640, China; qlpcirc@163.com (Q.L.); zhgluo@scut.edu.cn (Z.L.)
* Correspondence: xiaozhou7@163.com (X.X.); qygao@scut.edu.cn (Q.G.)

**Abstract:** Cross-linked esterified starch (CES) was prepared using a one-step method, where maize starch was selected as the raw material, sodium trimetaphosphate as the cross-linking agent, and acetic anhydride as the esterifying agent, respectively. A response surface experiment was systematically conducted for analyzing the correlation of the experimental variables (cross-linked temperature, pH, reaction time, sodium trimetaphosphate and acetic anhydride dosage) and properties of the product (peak and final viscosity). The Brabender viscosity, freeze-thaw stability, shearing resistance, and acid tolerance of the cross-linked acetylated dual modified starch were studied under different conditions of crosslinking degree and acetyl content. Meanwhile, the granular structure and morphology of the modified starch were analyzed. The results indicated that: after cross-linked acetylated dual modification, the starch had a distinct birefringence and granular structure, along with the creation of new carbonyl groups. The low degree of crosslinking and high acetyl contents were beneficial to the viscosity, which was significantly increased at both low and high temperatures. Moreover, the freeze-thaw stability of CES was elevated sharply after five cycles. In addition, CES displayed increased shear and acid tolerance compared to the original waxy maize, and their lowest differences between waxy maize and CES were only 0.62% and 0.59%, respectively. In summary, a novel method for starch modification was provided, and the synthesized CES was suggested to have exceptional performance for the food industry.

**Keywords:** maize starch; cross-linked esterified starch; freeze-thaw stability; shearing tolerance





## 1. Introduction

Waxy maize, which has a strong resistance to aging and outstanding pasting property due to the high content of amylopectin (over 95%), has been widely used in the food industry. However, the low thermal stability and acid tolerance of the unprocessed waxy maize starch restricted its commercial application in larger scales [1]. Therefore, efforts have been made to modify waxy maize starch for the improvement of these properties using physical and biological methods [2,3].

With the development of the food industry and the need for the special foods, modified starches with enhanced performances (e.g., freeze-thaw stability, shearing, and acid tolerance) are becoming more and more popular as food additives. For example, thermal stability is essential in sterilization during food packaging, and tolerance to shear force is prerequisite for mechanical agitation [4]. Moreover, for food preservation processes, the freeze-thaw tolerance of starch is crucial for food flavor [5,6]. Specifically, food in acid condition requires starch tolerant to acid [7,8]. Therefore, appropriate modification of waxy maize starch is aimed to extend its applications [9]. Up to now, several modified starches have been reported to obtain special properties, such as water-retention, emulsification, and film formation. These modifications significantly simplified food processing and increased the quality of the food [10–12].

Cross-linked esterified starch (CES) is a kind of modified starch produced via cross-linking polymerization and acetylation. CES has exceptional properties of thermal stability, along with acid and shearing tolerance. Therefore, CES was regarded as a promising material for the preparation of hydrogels. In the food industry, CES imparts the thickening needed in food applications, and it has been used for its stability and resistance to retrogradation [12]. More importantly, it exhibits a lower gelatinization temperature and a high grade of transparency [13,14]. Currently, the production of CES from potato, tapioca, ipomoea, and maize have been reported, while its one-step production from waxy maize starch is rarely investigated [15,16]. Compared to the conventional methods for CES production, the one-step synthesis of CES is a faster process which requires fewer reaction steps, which indicates that it is an economical method for industrial production. Therefore, the aim of this study was to produce CES from waxy maize starch though a one-step reaction. Moreover, assessments of CES's properties at various aspects were conducted. This study provided an important foundation for the industrial production of CES.

## 2. Materials and Methods

### 2.1. Agents

Waxy maize starch was purchased from Lihua Co., ltd. from Qinhuangdao, China. Food-grade acetic anhydride and sodium trimetaphosphate were bought from Chemical Agents of Guangzhou, China. The other agents and materials were all the products of Guangfu Co., ltd. from Tianjin, China; these chemicals were analytical agents used to ensure the reproducibility of these experiments.

### 2.2. Production of CES

Starch acetate was prepared as a 40% emulsion before the cross-link reaction. After completion of the chemical reaction, the pH value was adjusted to 8.0. Next, acetic anhydride was added to complete the esterification. During the esterification process, 5% of NaOH was provided to maintain the pH value at 8.0. After the esterification, the pH of the solution was adjusted to 6.5 by the addition of 0.5 mol/L hydrochloric acid. Subsequently, the insoluble substances in the mixture were filtered through a filter membrane, followed by drying at 45 °C in an incubator. The remaining substances were then screened by a molecular sieve of 100 mesh. Response surface method was applied to optimize the production process, and the viscosity was set as the outcome for evaluation. The experimental design is listed in Table 1.

**Table 1.** Preparation conditions, acetyl content, and bound phosphorus content of CES.

| Sample | Cross-Linking Temperature (°C) | pH | Cross-Linking Agent (%) | Reaction Time (h) | Acetic Anhydride (%) | Esterification Time (h) | Esterification Temperature (°C) | Acetyl Radical (%) | Substitution of Esterification (DS) | Combined Phosphorus (%) |
|---|---|---|---|---|---|---|---|---|---|---|
| Waxy maize starch | 0 | 0 | 0 | 0 | 0 | 1.5 | 25 | 0 | 0 | 0.007 |
| CLAC-1 | 45 | 10 | 0.7 | 1 | 3 | 1.5 | 25 | 0.89 | 0.034 | 0.017 |
| CLAC-2 | 40 | 10.5 | 0.9 | 2 | 3 | 1.5 | 25 | 0.83 | 0.032 | 0.025 |
| CLAC-3 | 40 | 10 | 0.7 | 2 | 3 | 1.5 | 25 | 1.02 | 0.039 | 0.022 |
| CLAC-4 | 35 | 10.5 | 0.7 | 2 | 3 | 1.5 | 25 | 1.17 | 0.045 | 0.018 |
| CLAC-5 | 40 | 10 | 0.9 | 3 | 3 | 1.5 | 25 | 0.87 | 0.033 | 0.025 |

### 2.3. Determination of Phosphorus Content in CES

Each 0.5 g sample was taken into the digestion bottle, and 15 mL of digested solution (sulfuric acid and nitric acid at 1:1) were added and mixed with the sample. Then, the bottle was heated until the color of the inner gas became white and the liquid was clarified. To reduce the side effect of the remaining nitric acid on the following reaction, 10 mL of sterilized water were added and the mixture was heated until the color of the inner gas became white. After cooling for 1 h, 45 mL of sterilized water was added into the system and the pH was adjusted to 7.0 with 10 mol/L NaOH. Finally, the solution was

transferred into a 100 mL volumetric bottle. After supply of water was adjusted to maintain the prescribed volume, the solution was preserved for further analysis.

The next step was as follows: a 25 mL sample was taken into a 50 mL volumetric bottle, and 4 mL of ammonium molybdate and 2 mL ascorbic acid (50 g/L) were added into the bottle. Subsequently, the bottle was boiled for 10 min and water was added to maintain the volume of the bottle after cooling down. The absorbance of the solution at 825 nm was determined, and the standard curve was plotted for analysis. The content of combined phosphorus was calculated as the ratio, according to Formula (1).

$$P(\%) = \frac{m_1 \times v_1 \times 100}{m_0 \times v_2 \times 10^6} \tag{1}$$

where $P$ represented the content of combined phosphorus in the sample. The symbols $v_1$ and $v_2$ represented the dilution volume and the final volume used for determination of absorbance. The symbols $m_1$ and $m_0$ represented the amount of the sample used in in the beginning and the amount of calculated phosphorus.

### 2.4. Determination of Acetyl Radical Content in CES

A total of 5 g of CES was weighed and dissolved in 50 mL of sterilized water in a 250 mL iodine flask. Samples were mixed with phenolphthalein in the solution. Then the amount of acetyl radical was analyzed by titration. After the reaction was finished, 25 mL of 0.5 mol/L NaOH was added into the solution, and it was cultivated at room temperature for 60 min for another titration. The second titration was conducted with 0.5 mol/L standard HCl, and the HCl volume was recorded as $V_1$ (mL). The control experiment with the original starch solution was applied, and the volume was recorded as $V_2$ (mL). The maximum degree of substitution (*DS*) of CES was 3, and Formulas (2) and (3) were used for the calculation.

$$W = \frac{(V_2 - V_1) \times 10^{-3} \times C \times 0.043}{m} \times 100\% \tag{2}$$

$$DS = \frac{162 \times W}{[4300 - 42W]} \tag{3}$$

where $W$ represented the content of acetyl radical, and $m$ represented the amount of the sample (g). $C$ represented the concentration of HCl (mol/L). The parameter 0.043 equals the amount of acetyl radical with 1 mL of HCl standard solution (1 mol/L).

### 2.5. Scanning Electron Microscope Analysis

Samples were taken using double conductive adhesive on the operation desk. These samples were then metal sprayed under vacuum. The samples were then plated on the specimen stage, and the picture was obtained from an SEM at the amplification rate of 3000.

### 2.6. Fourier Transform Infrared Spectroscopy (FT-IR) Analysis

Each 2 mg sample of CES was mixed with dried KBr for FT-IR analysis. The mixture was forming as a tablet, followed by analysis. The resolution was set at 2 cm$^{-1}$ and scanning was performed from 4000 to 400 cm$^{-1}$.

### 2.7. Determination of Brabender Viscosity Curve

A total of 6.00 g of the dried sample was taken into the Brabender rotational viscometer, and enough water was added to dilute the sample to a 5% concentration (*w/w*). The torque was set at 350 cmg, and the agitation speed was set at 250 rpm. The starting temperature was 30 °C and it was advanced at the speed of 1.5 °C/min. After reaching 95 °C, the system was cooled down to 50 °C at room temperature and then incubated for 30 min. The changes of the parameters were recorded and the Brabender viscosity curve was obtained.

*2.8. Freeze-Thaw Stability Assay*

A prescribed amount of CES was taken into the shake flask and 250 mL water was added to reach the final concentration of 6% ($w/w$). The solution was divided into 5 portions and the mass was measured. The five bottles were then held at $-15\ ^\circ$C for 24 h, followed by thawing at room temperature. Subsequently, samples were centrifuged at 3000 rpm for 20 min to remove the supernatant. The mass of the remaining tube, together with the sample, were measured. The freeze-thaw cycle was conducted 5 times, and the water separating proportion was calculated with Formula (4)

$$X = \frac{m_2 - m_3}{m_2 - m_1} \times 100\% \tag{4}$$

where $X$ represented the total water separating proportion of the sample. Parameter $m_1$ represented the mass of the centrifugation tube, and $m_2$ was the total mass of the tube and the starch. Parameter $m_3$ represented the total amount of remaining starch and the tube after centrifugation.

*2.9. Shearing Tolerance and Acid Tolerance of CES*

Samples of CES were diluted in water to reach the final concentration of 3%. The temperature of the system was advanced to 95 $^\circ$C at the rate of 1.5 $^\circ$C/min. The system was then cooled down to 30 $^\circ$C for the analysis of shear tolerance. The viscosity of the sample was determined after shearing at 10,000 r/min for 15 min, and the relative viscosity was calculated to evaluate the shear tolerance of the CES.

The acid tolerance of CES was conducted in a similar way. After cooling down, the pH of the 3% CES samples was adjusted to 2 with acetic acid. After stirring for 10 min, the viscosity was determined, and the change of viscosity was applied to evaluate the acid tolerance of CES.

## 3. Results and Discussion

*3.1. SEM Analysis of the Morphology of CES*

The CES samples were scanned in the SEM to check the morphology of the samples (Figure 1). The results indicated that the CES was successfully synthesized, and the particles were neatly arranged. The CES were polygonal particles with clear edges and smooth surfaces. More importantly, the particles had similar sizes in the visual field. Compared to the waxy maize starch, the morphology of CES was rarely changed. Different from the other methods of starch modifications [17,18], the one-step synthesis of CES has a limited effect on morphology. This suggested that the one-step synthesized CES had biocompatibility with currently used starch, showing considerable flexibility in food applications.

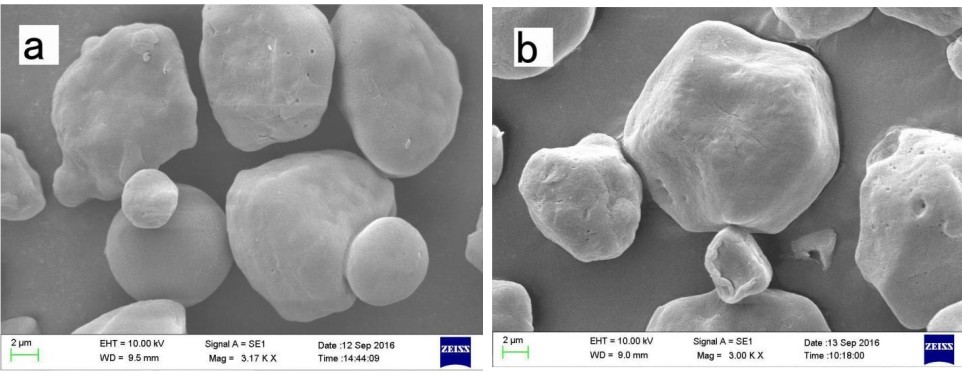

**Figure 1.** *Cont.*

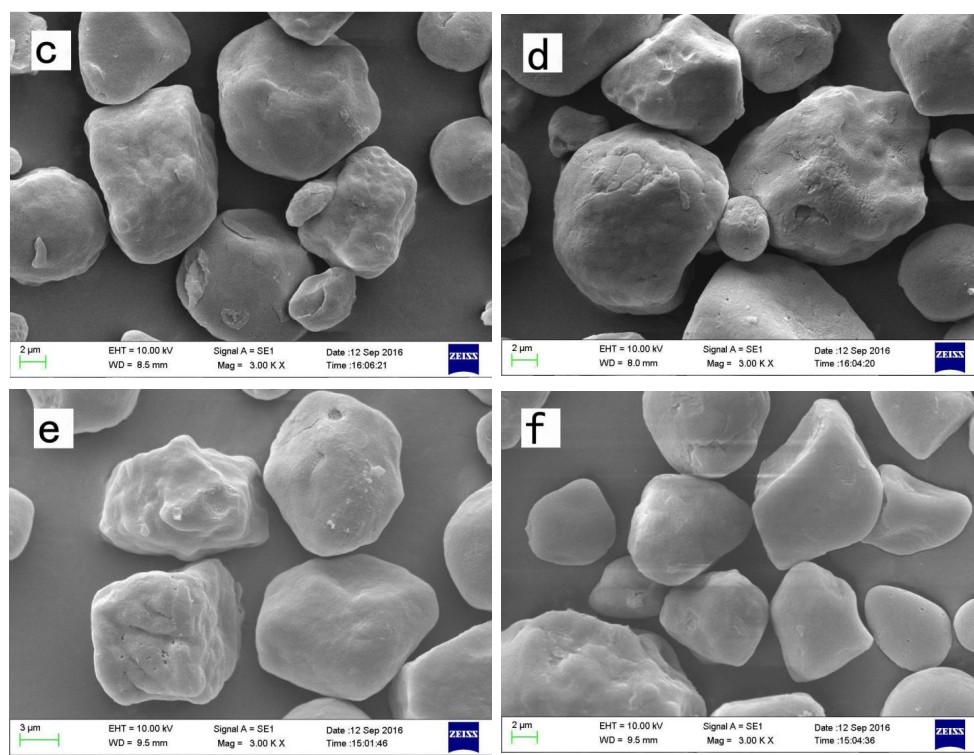

**Figure 1.** SEM (**a**) raw waxy corn; (**b**) CLAC-1; (**c**) CLAC-2; (**d**) CLAC-3; (**e**) CLAC-4; (**f**) CLAC-5.

### 3.2. FT-IR Analysis of CES

The FT-IR analysis of CES was conducted to estimate the exact changes in chemical properties (Figure 2). The significant changes of CES in FT-IR analysis were located at 1734 cm$^{-1}$ and 1247 cm$^{-1}$, which were the characteristic absorbance peaks of starch acetate. Therefore, these results demonstrated that acetate was successfully cross-linked with starch, namely, the peak at 1734 cm−1 represented the vibration of C=O, and the peak at 1247 cm$^{-1}$ represented the vibration of C-O. As the elevated esterification ratio of starch, the stronger absorbance was detected at 1247 cm$^{-1}$, along with enhanced absorbance, also at 1734 cm$^{-1}$. Moreover, since a relationship existed between the DS value and the peak, the introduction of the acetyl radical group during the synthesis of CES also contributed to the promotion of the esterification process. However, the characteristic peak of cross-inked starch was not detected. The reason was ascribed to the low degree of cross-linking of CES [17].

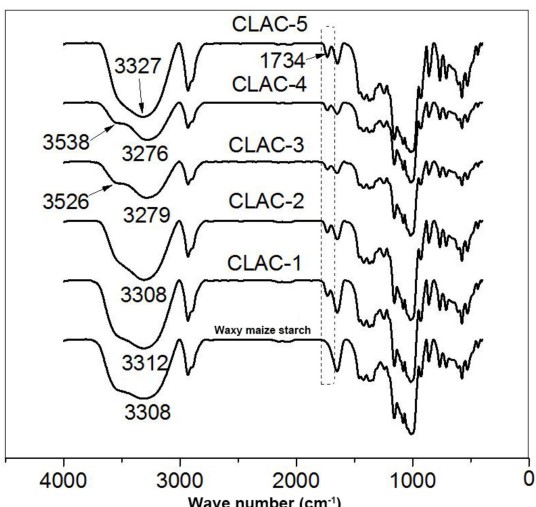

**Figure 2.** Infrared spectrum of the cross−linked esterified starch sample.

### 3.3. Brabender Viscosity Curve of CES

Next, the Brabender viscosity curve of the synthesized CES was determined, since it is an important characteristic of functional starch (Figure 3). Due to the different degrees of cross-linking, the viscosities of CES were increased at the initial phase, and then decreased if the phosphorus content was at a high level. The reason was deduced as the changes in the hydrogen bond energy within the cross-linked starch. The hydrogen bond of starch will be reduced with increased temperature, leading to the expansion of the sample and a sharply enlarged molecular chain, which will affect the movement of the starch molecules. However, once the threshold is reached, the expansion of the starch molecules will be terminated, accompanied by the reduced viscosity of the CES.

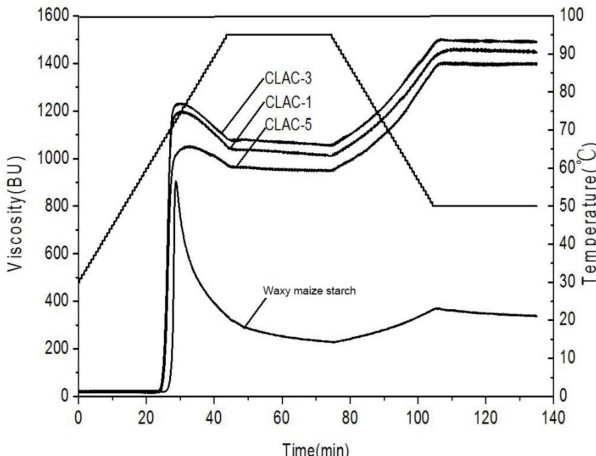

**Figure 3.** Brabender viscosity curve of CES.

In parallel, the Brabender viscosity curves were also determined using the same level of cross-linking, but different degrees of esterification (Figure 4). As a result, with the increased content of acetyl acetate, the viscosities of CES were increased at both low and high temperatures. The polar property of the acetyl acetate group at the side chain of the starch led to the complexity of the structure, while ultimately elevating the viscosity of CES [17,19]. The cross-linking of starch is believed to impart viscosity stability and a desired short-textured property of the paste [18]. However, for each application, there is an optimum level and balance between hydroxypropyl substitution and cross-linking. Therefore, the different synthesis conditions of one-step cross-linking in our study provided a wide range of viscosity for extended applications.

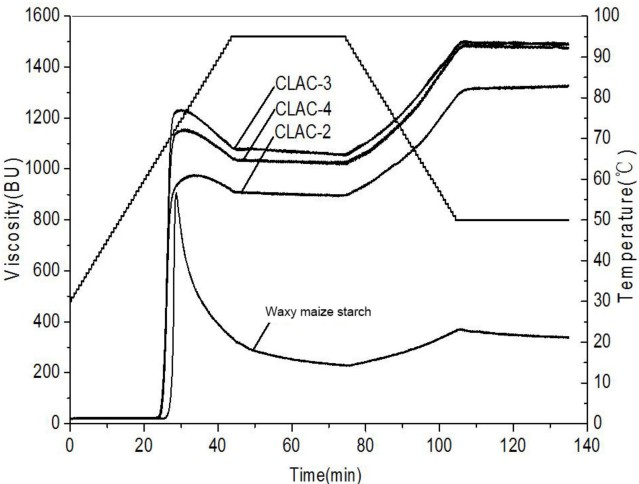

**Figure 4.** Brabender viscosity curve CES.

### 3.4. Determination of Freeze-Thaw Stability of CES

The freeze-thaw stability of CES was conducted (Table 2). Compared to the original waxy maize starch, the sweating rate of CES was reduced with the increased time of freezing-thawing. Particularly, there was no water sweated in the initial cycle of the samples CLAC-1, CLAC-2, and CLAC-5. After 5 cycles of freezing-thawing, the sweating rate was significantly reduced. As seen in the SEM analysis, CES was shown to be smoother for samples with a higher concentration of acetyl acetate, indicating that low cross-linking and a high degree of acetyl were better for controlling the molecular interaction of starch. Moreover, varied freezing and thawing points suggested that different water contents between these samples contributed to the stability of the starch gelatinization system, which also resulted in the promotion of stability in the freeze-thaw cycles. Modifications and starch concentrations were believed to be important to the stability of the starch [20]. The ice crystals were formed in the starch gel during freezing and became larger throughout the freeze–thaw cycles. When these crystals melted, the rough pores may have acted as a sponge, influencing the retention capacity for water [21]. Modification of starch may contribute to the formation of a sponge-like structure, resulting in high freeze-thaw stability.

**Table 2.** Freeze-thaw stability of waxy native corn starch and cross-linked esterified composite modified starch paste.

| Sample | Content of Acetyl Acetate (%) | (DS) | Content of Conjoined Phosphate (%) | Sweating Rate (%) | | |
|---|---|---|---|---|---|---|
| | | | | 1 Time | 3 Times | 5 Times |
| Waxy maize starch | 0 | 0 | 0.007 | 45.23 | — | — |
| CLAC-1 | 0.89 | 0.034 | 0.017 | 0.14 | 31.03 | 35.52 |
| CLAC-2 | 0.83 | 0.032 | 0.025 | 24.18 | 28.58 | 43.40 |
| CLAC-3 | 1.02 | 0.039 | 0.022 | 5.56 | 34.03 | 48.52 |
| CLAC-4 | 1.17 | 0.045 | 0.018 | 19.52 | 26.62 | 44.65 |
| CLAC-5 | 0.87 | 0.033 | 0.025 | 5.93 | 29.02 | 34.42 |

### 3.5. Acid Tolerance and Shearing Tolerance of CES

The tolerance of CES to shearing was determined (Table 3), and the results indicated the increased shearing tolerance of CES compared to the original waxy maize starch. For the waxy maize starch, the starch molecules will be separated after continuous shearing. In contrast, increasing the interaction between starch molecules by cross-linking contributed to the increased viscosity of CES, making it less sensitive to mechanical shearing. For the samples with different degrees of cross-linking, higher degrees of linking contributed to better tolerance to shearing. In contrast, the degree of esterification had a limited impact on the viscosity of CES. In summary, the shearing tolerance was closely related to the degree of cross-linking, and CLAC-2 and CLAC-5 displayed the highest tolerance to mechanical shearing.

**Table 3.** Shear tolerance and acid tolerance of waxy native corn starch and cross-linked esterified composite modified starch paste.

| Sample | Initial Viscosity (mpa·s) | After Shearing (mpa·s) | Difference after Shearing (mpa·s) | Relative Difference after Shearing (%) | After Acidification (mpa·s) | Difference after Acidification (mpa·s) | Relative Difference after Acidification (%) |
|---|---|---|---|---|---|---|---|
| Waxy maize starch | 267 | 26 | 241 | 0.90 | 312 | 45 | 16.85 |
| CLAC-1 | 904 | 78 | 826 | 0.91 | 897 | 7 | 0.77 |
| CLAC-2 | 507 | 193 | 314 | 0.62 | 504 | 3 | 0.59 |
| CLAC-3 | 873 | 87 | 786 | 0.90 | 862 | 11 | 1.26 |
| CLAC-4 | 662 | 131 | 531 | 0.80 | 654 | 8 | 1.21 |
| CLAC-5 | 593 | 189 | 404 | 0.68 | 580 | 13 | 2.19 |

The acid tolerance of CES was also increased when compared to waxy maize starch. When the pH value was set at 2, the viscosity of CES was not changed, while the results for the waxy maize starch showed the opposite effect. An acidic condition can accelerate the release of starch particles, which then reduced the viscosity of the starch. In contrast,

the CES exhibited better performance in an acidic condition, which indicated that it had promising applications as a food additive in acidic conditions [22,23].

### 4. Conclusions

In this work, the one-step synthesis of CES was conducted to increase the performance of functional starch. These data demonstrated that the synthesized CES displayed increased stability in the freeze-thaw experiment, and the tolerance to shearing and acid were also significantly enhanced, showing promise for future applications. At the same time, the morphology of CES was insignificantly changed, showing its biocompatibility for current food systems. More importantly, the single-step synthesis process is an economical method for large-scale production, which will facilitate the extended application of CES.

**Author Contributions:** Q.G. and Z.L. designed the research; X.X. performed the experiments; Q.L. and X.X. analyzed the data; and X.X. wrote the manuscript. All authors have read and agreed to the published version of the manuscript.

**Funding:** This work was financially supported by the International Cooperation Project of Guangdong (2019A050510005), the Research Project of Guangzhou (202102020621), and the Fundamental Research Project of College (2020ZYGXZR037).

**Institutional Review Board Statement:** Not applicable.

**Informed Consent Statement:** Not applicable.

**Data Availability Statement:** Not applicable.

**Conflicts of Interest:** The authors declare no conflict of interest.

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
