# Peer review of "One-Step Synthesis of Cross-Linked Esterified Starch and Its Properties"

_applsci, doi:10.3390/app12084075_

Round 1

Reviewer 1 Report

Dear Editor, dear Authors,

I had the opportunity to review the paper entitled „One-step synthesis of cross-linked esterified starch and its properties“. After I checked all section of this paper, I believe that authors bring a new, scientific-relevant procedure, so I suggest only minor revision.

There are my comments for this paper:

  • it is necessery to improve English through the whole manuscript;
  • avoid using „we“ through the whole manuscript;
  • reorganize the abstract in order to facilitate reading and put more obtained results here;
  • Table 1 - uniform units’ representation (some are with “/”, while some with “()”
  • line 75: replace “next” with “the further step was involved….”
  • line 87: uniform both “m” (one m is in all caps);
  • 3: please, remove square brackets, use normal one;
  • Provide better quality of each figure, and try to uniform font with the mdpi template;
  • line 222-227 please, reorganize this part in order to facilitate reading;
  • please reorganize the Conclusion and present in better manner the highlight of this research, and expand this part with one more section to explain the obtained results;

Reviewer 2 Report

  • The abstract is not written well. It must show some specific data from the research. In the current form, it seems like an introduction. Please rewrite the abstract with more details about the results and methodologies.
  • Please suggest some applications for CES at the industrial level.
  • Compared with other methodologies of CES production, what is the advantage of one-step production?
  • Line 30: the sentence is not suitable for starting a paragraph. It needs some paraphrasing: “With the development of food industry and the in-depth research of food processing technology, high quality of starch becomes more and more important as an adjuvant for different kinds of food”.
  • According to this sentence, the concentration of starch emulsion is too high, after the addition of a crosslinker, it becomes hard and condenses. Is it a suitable concentration for the starch emulsion or less than this such as 20% emulsion is also suitable? “Starch acetate was prepared as 40% emulsion before the cross-link reaction”.
  • The experimental results need to be compared with other studies. The explanation of the results is very short and must discuss more.
  • As a suggestion, the authors can include a future prospect at the end of this research.
  • There are some old references (before 2010) that can be replaced with newly published papers.
  • The following reference can be used in the manuscript: Sabbagh, F., & Kim, B. S. (2022). Recent advances in polymeric transdermal drug delivery systems. Journal of Controlled Release341, 132-146.
